# Epigenetic Modulation of Radiation-Induced Diacylglycerol Kinase Alpha Expression Prevents Pro-Fibrotic Fibroblast Response

**DOI:** 10.3390/cancers13102455

**Published:** 2021-05-18

**Authors:** Chun-Shan Liu, Reka Toth, Ali Bakr, Ashish Goyal, Md Saiful Islam, Kersten Breuer, Anand Mayakonda, Yu-Yu Lin, Peter Stepper, Tomasz P. Jurkowski, Marlon R. Veldwijk, Elena Sperk, Carsten Herskind, Pavlo Lutsik, Dieter Weichenhan, Christoph Plass, Peter Schmezer, Odilia Popanda

**Affiliations:** 1Division of Cancer Epigenomics, German Cancer Research Center (DKFZ), 69120 Heidelberg, Germany; chun-shan.liu@dkfz-heidelberg.de (C.-S.L.); r.toth@dkfz-heidelberg.de (R.T.); a.bakr@dkfz-heidelberg.de (A.B.); a.goyal@dkfz-heidelberg.de (A.G.); mohammedsaiful.islam@dkfz-heidelberg.de (M.S.I.); k.breuer@dkfz-heidelberg.de (K.B.); a.mayakonda@dkfz-heidelberg.de (A.M.); yu-yu.lin@dkfz-heidelberg.de (Y.-Y.L.); p.lutsik@dkfz-heidelberg.de (P.L.); d.weichenhan@dkfz-heidelberg.de (D.W.); c.plass@dkfz-heidelberg.de (C.P.); p.schmezer@dkfz-heidelberg.de (P.S.); 2Division of Molecular Thoracic Oncology, German Cancer Research Center (DKFZ), 69120 Heidelberg, Germany; 3Institute of Biochemistry and Technical Biochemistry, University of Stuttgart, 70569 Stuttgart, Germany; pstepper@cemm.oeaw.ac.at (P.S.); JurkowskiT@cardiff.ac.uk (T.P.J.); 4Department of Radiation Oncology, Universitätsmedizin Mannheim, Medical Faculty Mannheim, Heidelberg University, 68167 Mannheim, Germany; Marlon.Veldwijk@medma.uni-heidelberg.de (M.R.V.); Elena.Sperk@umm.de (E.S.); Carsten.Herskind@medma.uni-heidelberg.de (C.H.)

**Keywords:** bromodomain inhibitors, DNA methylation, EZH2 inhibitors, radiotherapy-induced fibrosis

## Abstract

**Simple Summary:**

To reduce long-term fibrosis risk after radiotherapy, we demonstrated with different experimental approaches that modulation of the epigenetic pattern at the *DGKA* enhancer can attenuate pro-fibrotic reactions in human fibroblasts. We used (epi)genomic editing of the *DGKA* enhancer and administration of various epigenetic drugs and were able to modulate radiation-induced expression of *DGKA* and pro-fibrotic collagens. Based on our results, clinical application of bromodomain inhibitors will open promising ways to epigenetically modulate *DGKA* expression and might provide novel therapeutic options to prevent or even reverse radiotherapy-induced fibrotic reactions.

**Abstract:**

Radiotherapy, a common component in cancer treatment, can induce adverse effects including fibrosis in co-irradiated tissues. We previously showed that differential DNA methylation at an enhancer of diacylglycerol kinase alpha (*DGKA*) in normal dermal fibroblasts is associated with radiation-induced fibrosis. After irradiation, the transcription factor EGR1 is induced and binds to the hypomethylated enhancer, leading to increased *DGKA* and pro-fibrotic marker expression. We now modulated this *DGKA* induction by targeted epigenomic and genomic editing of the *DGKA* enhancer and administering epigenetic drugs. Targeted DNA demethylation of the *DGKA* enhancer in HEK293T cells resulted in enrichment of enhancer-related histone activation marks and radiation-induced *DGKA* expression. Mutations of the EGR1-binding motifs decreased radiation-induced *DGKA* expression in BJ fibroblasts and caused dysregulation of multiple fibrosis-related pathways. EZH2 inhibitors (GSK126, EPZ6438) did not change radiation-induced *DGKA* increase. Bromodomain inhibitors (CBP30, JQ1) suppressed radiation-induced *DGKA* and pro-fibrotic marker expression. Similar drug effects were observed in donor-derived fibroblasts with low DNA methylation. Overall, epigenomic manipulation of *DGKA* expression may offer novel options for a personalized treatment to prevent or attenuate radiotherapy-induced fibrosis.

## 1. Introduction

Radiation is a powerful therapeutic approach to limit tumor growth. More than 50% of all cancer patients receive radiotherapy at some point during treatment for curative or palliative purposes [1,2]. However, as radiation also injures the co-irradiated, tumor-surrounding normal tissue, side effects may occur either early (up to three months) or late (after months or years) after therapy [3,4]. A late side effect is subcutaneous fibrosis, which can substantially reduce quality of life and may lead to severe functional defects. In breast cancer patients, about 21% of survivors developed subcutaneous breast fibrosis eight years after radiotherapy with an intraoperative boost and external beam whole-breast irradiation [3]. In addition, around 68% of head and neck cancer patients showed mild-to-severe neck fibrosis after radiotherapy with an increasing risk every year after therapy [4]. Radiation-induced tissue damage can trigger processes similar to wound healing, which initiate tissue regeneration but, if not attenuated in time, result in scarring and fibrosis. On a molecular level, fibrosis includes fibroblast-to-myofibroblast trans-differentiation and increased production of extracellular matrix (ECM) proteins such as collagens causing indurations and scars [5,6]. Although the molecular understanding of fibrosis is steadily improving, the question how to prevent or treat radiation-induced fibrosis remains an important issue to improve quality of life of cancer survivors.

Epigenetic mechanisms such as DNA methylation and histone modifications are major contributors to fibrosis development by controlling the cellular transcriptome [7,8,9,10,11]. We recently described an epigenetic predisposition associated with radiation-induced fibrosis risk [12]. This risk was controlled by the DNA methylation status of an intronic enhancer element of the *diacylglycerol kinase alpha* (*DGKA*) gene. The DGKA kinase phosphorylates the lipid messenger diacylglycerol converting it to phosphatidic acid, and therefore participates in lipid signaling, exosome production, cell migration, cell proliferation, and immune response functions [13,14], making it an important candidate involved in radiation-induced fibrosis. The *DGKA* enhancer includes two binding sites for early growth response protein 1 (EGR1), a radiation-inducible transcription factor (TF) and stress response regulator [12,15]. DNA methylation levels at this region were hypothesized to be responsible for radiation-induced *DGKA* expression and activation of pro-fibrotic marker proteins like collagen and other ECM proteins [12]. In fibroblasts with a hypermethylated *DGKA* enhancer, EGR1 was not able to bind, thus limiting *DGKA* transcription and ECM production (Figure 1A). Modulating the chromatin structure by epigenetic editing or small molecules affecting epigenetic players offers an attractive way to attenuate fibrosis risk in radiotherapy patients.

Chromatin structure is an important factor in regulating transcriptional activity with histone methylation and acetylation as the major modifications. Thus, histone-modifying enzymes are essential to control gene expression [16]. Relevant candidates to inhibit the radiation-induced fibrotic response could be histone methyltransferase, like EZH2, or histone acetyltransferase (HAT) inhibitors. EZH2 is part of the polycomb repressive complex 2 (PRC2), which mediates successive methylation of lysine 27 on histone H3 up to trimethylation (H3K27me3), causing chromatin condensation and transcriptional repression, but it can also interact with other proteins to co-activate PRC2-independent gene transcription [17,18]. EZH2 was shown to promote differentiation of fibroblasts to myofibroblasts in idiopathic pulmonary fibrosis [19] and inhibition of EZH2 reduced fibrosis-related gene expression in atrial fibrosis [20], liver fibrosis [21], and systemic sclerosis [22]. HATs acetylate histones, which increases chromatin accessibility, thereby facilitating gene transcription [16]. Inhibition of CBP/p300, a HAT protein complex containing bromodomains, was suggested to contribute to cardiac and renal fibrosis in a murine model, and pharmacological inhibitors of these proteins have been described as antifibrotic agents [23]. Acetylation of histones was recognized and translated into transcription by bromo- and extra-terminal domain (BET) proteins and can be suppressed by the small molecule inhibitor JQ1 [24]. Recently, it was shown that the inhibition of BET proteins by JQ1 could counteract the pro-fibrotic transcriptional response in fibroblasts after bleomycin treatment [25], probably by affecting the epigenetic pattern at the *DGKA* enhancer.

The function of the epigenetic modifying enzymes in radiation-induced fibrosis has not been described yet and the option to manipulate fibrosis-relevant epigenetic patterns should be investigated in more detail. We therefore studied the impact of these epigenetic mechanisms on radiation-induced *DGKA* expression in cultured normal human dermal fibroblasts with different DNA methylation and histone patterns at the *DGKA* enhancer region. We directly manipulated the *DGKA* enhancer region by epigenetic or genetic editing. Thus, demethylation with dCas9 fused to a VPR-TET3 fusion protein, a demethylating enzyme fused to a transcriptional activation domain, and mutation of the EGR1-binding sites at the *DGKA* enhancer region by CRISPR/Cas9, both altered the induction of *DGKA* and pro-fibrotic markers after ionizing radiation (IR) as expected. Furthermore, we showed that the bromodomain inhibitors JQ1 and CBP30 attenuated radiation-induced *DGKA* and pro-fibrotic marker expression; this was not the case for the EZH2 inhibitors GSK126 and EPZ6438. Our results open up new therapeutic applications to prevent or attenuate radiotherapy-induced fibrosis.

## 2. Materials and Methods

### 2.1. Cell Lines

Human foreskin BJ fibroblasts (CRL-2522) and HEK293T cells (CRL-11268) were obtained from American Type Culture Collection (ATCC, Manassas, VA, USA). BJ cells were cultivated in minimal essential medium (MEM, ThermoFisher, Waltham, MA, USA) plus 10% fetal bovine serum (FBS, SigmaAldrich, St. Louis, MO, USA) and 1% penicillin-streptomycin (PenStrep, ThermoFisher, Waltham, MA, USA). BJ cells were used below passage 20 during the experiments. HEK293T cells were maintained in Dulbecco’s modified Eagle medium (DMEM, ThermoFisher, Waltham, MA, USA, containing 10% FBS and 1% PenStrep. Normal human dermal fibroblasts (NHDFs, L1-4 and H1-3) were established from female donors aged 47 to 84 years (median age, 66 years) at the University Medical Center Mannheim, Germany, as part of the EURATOM/ESTRO GENEPI project [26]. Fibroblasts were outgrown from skin biopsies, which were taken from the un-irradiated inner upper arm of donors. The study protocol was approved by the Ethics Committee of the University Hospital of Mannheim. All donors provided informed written consent. NHDFs were cultured in AmnioMAX C-100 Basal Medium (ThermoFisher) containing 7.5% AmnioMAX C-100 Supplement (ThermoFisher), 7.5% FBS, 2 mM Glutamin, and 1% PenStrep, and were used below passage 10 during the experiments. Cell lines were authenticated using the Multiplex Cell Authentication and Multiplex Cell Contamination Test by Multiplexion (Heidelberg, Germany) as described [27,28]. Profiles matched known profiles for BJ and HEK293T cells or were unique for the fibroblasts. No Mycoplasma, SMRV, or interspecies contaminations were detected.

### 2.2. Preparation of Isogenic Cell Lines with Reduced DNA Methylation at the DGKA Enhancer

For targeted demethylation, four single guide RNAs (sgRNAs, Appendix A) flanking the two EGR1-binding sites at the *DGKA* enhancer were designed using the CRISPR-MIT tool [29]. Assembly of the targeting vector was done in two steps. First, the U6-gRNA cassettes were assembled using Gibson assembly and cloned into an empty PiggyBac Cumate Switch Inducible Vector (PBQM, System Biosciences, Palo Alto, CA, USA) between the PuroR gene and the 3′ TR using Cas9/CRISPR cleavage and Gibson cloning of U6-gRNAs cassettes. In the second step, the dCas9-VPR-Tet3CDdel1 gene was inserted from a pENTR plasmid, behind the cumate modulated promoter using the Gateway LR Clonase II enzyme mix (ThermoFisher, Waltham, MA, USA). The final construct was verified with Sanger sequencing. HEK293T cells were co-transfected with this vector and the Super PiggyBac Transposase expression vector (System Biosciences, Palo Alto, CA, USA) by using TransIT-LT1 (Mirus Bio, Madison, WI, USA) according to the manufacturer’s protocol, and selected by puromycin (2 µg/mL). Cumate (300 µg/mL) was supplemented to the cell culture medium to induce dCas9-VPR-TET3 expression and cells were cultured over 2 passages before harvesting.

### 2.3. Preparation of Isogenic Cell Lines with Edited EGR1-Binding Sites at the DGKA Enhancer

Site-specific sgRNAs (sgRNAE1 and sgRNAE2, Appendix A) were designed as above and each was incorporated into the BsmBI site of the Lenti-CRISPR v2 plasmid vector (#52961, Addgene) [30]. The Lenti-CRISPR-sgRNAE1, -sgRNAE2, or -sgRNA-Gal plasmid was co-transfected with packaging plasmid psPAX2 and pMD2.G (both from PlasmidFactory GmbH & Co. KG, Bielefeld, Germany) into HEK293T cells by TransIT LT1 transfection reagent (Mirus Bio, Madison, WI, USA), and the medium was replaced with fresh medium the next day. The supernatant containing lentivirus carrying Lenti-CRISPR-sgRNAE1, -sgRNAE2, or -sgRNA-Gal was harvested 2 days after transfection. Pooled sable clones (sgE1 and sgE2) with edited EGR1-binding motifs at the *DGKA* enhancer were generated by transduction with lentivirus carrying Lenti-CRISPR-sgRNAE1 or -sgRNAE2 in BJ cells and followed by puromycin (2 µg/mL) selection. Control cells (sgC) were infected with lentivirus carrying Lenti-CRISPR-sgRNA-Gal followed by puromycin selection. Edited sequences were verified by Sanger sequencing (GATC Biotech AG, Konstanz, Germany). The primers used for PCR and sequencing were: For: 5’-CCCCAAGTCACACAGTGGTAT-3’ and Rev: 5’-CGAGACCTTGCACAATGCAG-3’. BJ cells and BJ-based stable cell pools (sgC, sgE1, and sgE2) were used below passage 20 during the experiments.

### 2.4. Radiation and Drug Treatment

Cells were irradiated using the^137^Cs Gammacell 40 Exactor (Best Theratronics, Ottawa, Ontario, Canada) at 1 Gy/min for the indicated doses. After irradiation, cells were kept in culture for the described treatment. If not otherwise indicated, cells were pretreated with (+)JQ1 (BPS Bioscience, San Diego, CA, USA), CBP30 (SigmaAldrich, St. Louis, MO, USA), GSK126 (Cat#15415, Cayman Chemical Co., Ann Arbor, MI, USA.), EPZ6438 (LKT-E6397, LKT Laboratories, Inc, St. Paul MN, USA), or vehicle (DMSO) for 48 h, irradiated with 6 Gy, and incubated for additional time with concurrent drug or vehicle treatment. Dimethyl sulfoxide (DMSO) was used to dissolve all compounds and used as vehicle control at equal volumes as for the drug treatment. Maximal DMSO concentration in assays was 0.1%.

### 2.5. siRNA Knockdown

All siGENOME upgrade siRNAs (Appendix A) were obtained from Horizon Discovery. A set of 4 siGENOME upgrade siRNAs was pooled together at a final concentration of 20 μM. Before transfection, 1 × 10^5^ BJ cells were cultivated in 3.5 cm dishes for 16–18 h. Cells were transfected using 2 μL of Lipofectamine DharmaFECT1 transfection reagent (#T-2001-03, Horizon Discovery, Cambridge, UK) for 0.16 pmol pooled siRNA yielding a final concentration of 80 nM siRNA in the assays. Cells were used for irradiation after 48 h.

### 2.6. Cell Viability Assay

To measure cell viability, the metabolic capacity of the cells was analyzed by CellTiter Blue reagent (#G8081, Promega, Madison, WI, USA). 3 × 10^3^ BJ cells in 100 μL medium per well were cultivated in 96-well plates and treated as described. 20 μL CellTiter Blue Reagent was added to 100 µL of cells, incubated for 2 h, and the fluorescence intensity was measured at a 560 nm excitation wavelength and a 590 nm emission wavelength using a SpectraMax M5 plate reader.

### 2.7. Quantitative DNA Methylation Analysis

DNA methylation at the *DGKA* enhancer locus was quantified using EpiTYPER MassARRAY technology (Agena Bioscience, San Diego, CA, USA). In brief, 1 µg genomic DNA was bisulfite-converted using the EZ DNA methylation kit (Zymo Research, Irvine, CA, USA). Regions of interest were amplified from bisulfite-treated DNA by Q5 DNA Polymerase (#M0491S, New England Biolabs, Ipswich, MA, USA) using specific primers, which were designed by using EpiDesigner software (Agena Bioscience, San Diego, CA, USA). The primers used in this study were: Epi_DGKA_F: 5′- GATTGGGAAATATTAGATTTGTTGG-3′ and Epi_DGKA_R: 5′- TTCCTAACCATAACCCCATTTTATT-3′. Methylation was quantified with EpiTYPER software version 1.2.

### 2.8. RNA Isolation and RNA Quantification by Real-Time PCR

RNA was isolated using the RNeasy mini kit (#74106, Qiagen, Hilden, Germany). Complementary DNA was prepared using SuperScript III reverse transcriptase (#12574026, ThermoFisher, Waltham, MA, USA) and quantified by real-time PCR using a LightCycler 480 (Roche, Basel, Switzerland) and universal probe library hydrolysis probes. Relative gene expression was calculated by the 2^−ΔΔCT^ method [31]. Target gene expression was normalized to the average of two housekeeping genes, HPRT1 and GAPDH. Primers (Appendix A) were designed using the Universal Probe Library Assay Design Center application.

### 2.9. RNA Sequencing

RNA was isolated using the RNeasy Mini kit, and RNA integrity was analyzed by a 2100 Bioanalyzer using the RNA 600 Nano Kit according to the manufacturer’s protocol. Sequencing libraries were prepared by the Genomics and Proteomics Core Facility (DKFZ, Heidelberg, Germany) from total RNA (*n* = 4 biological replicates from sgC or sgE1 cells) using the Illumina TruSeq Stranded Total RNA Library Prep Kit according to the manufacturer’s instructions.

For sequencing, multiplexes were sequenced in a paired-end setting (100 bp) on an Illumina NovaSeq 6000 machine. Data was processed using the nfcore/rnaseq (version 1.2) [32] pipeline with the following options: --reverseStranded --pairedEnds. Reads were aligned to the hg38 reference genome using HISAT2 (version 2.1.0) [33]. Gene and transcript level read counts were assessed using Stringtie2 (version 1.3.4) and the accompanying prepDE.py script [34] based on Gencode v29 reference data. Further processing and statistical analysis was performed using DeSeq2 (version 2_1.28.1) [35] in R 4.0.2. Multidimensional scaling (MDS) was visualized using Glimma (1.16.0) [36]. Gene set enrichment analysis was done using topGO (2.40.0) [37] and visualized with GeneTonic (1.1.1) [38].

### 2.10. Protein Extraction and Western Blotting

For total protein extraction, cells were lysed in RIPA buffer (10 mM Tris pH 8.0, 1 mM EDTA, 0.5 mM EGTA, 1% Triton X-100, 0.1% Sodium Deoxycholate, 0.1% SDS, and 140 mM NaCl) supplemented with protease inhibitor cocktail (Roche, Basel, Switzerland). For nuclear protein extraction, cells were lysed and extracted by using NE-PER Nuclear and Cytoplasmic Extraction Reagents (ThermoFisher, Waltham, MA, USA) according to the manufacturer’s instructions. All protein concentrations were determined using a BCA assay kit (SigmaAldrich, St. Louis, MO, USA) and measured with the SpectraMax Pro 5.44 at a wavelength of 560 nm. Protein extracts were prepared in Laemmli buffer (BioRad, Hercules, CA, USA), separated on 4–15% Criterion TGX Precast Gels (BioRad, Hercules, CA, USA), and transferred to PVDF membrane. Membranes were blocked at room temperature (RT) for 1 h in 5% skimmed milk in TBST buffer (20 mM Tris pH8.0, 150 mM NaCl and 0.2% Tween-20), and incubated overnight with the indicated primary antibodies at 4 °C. The primary antibodies used in this study were: H3K27ac (1:1000, Active Motif, #39133), H3k27me3 (1:1000, CST9733 Cell Signaling, Danvers, MA, USA), histone 3 (ab1791 Abcam, Cambridge, UK), Cas9 (1:5000, C15310258 Diagenode, Seraing, Belgium), and beta-actin (1:3000, sc-47778HRP, Santa Cruz, Santa Cruz, CA, USA). Membranes were washed, incubated with HRP-conjugated secondary antibodies (1:3000, CST7074, Cell Signaling) at RT for 1 h. HRP signals were detected using Immobilon Western Chemiluminescent HRP Substrate (Merck-Millipore, Burlington, MA, USA) and measured with the Amersham imager 680 (GE Healthcare, Chicago, IL, USA). Quantification was performed using ImageJ software version 1.52a [39], and normalization was done using either the beta-actin or the two H3 bands as loading control.

### 2.11. Electrophoretic Mobility Shift Assay (EMSA)

EMSA was carried out using a LightShift Chemiluminescent EMSA Kit (ThermoFisher, Waltham, MA, USA) according to the manufacturer’s instructions. In brief, 1 µg/µL of EGR1-overexpressing cell lysate (OriGene, Rockville, MD, USA) was incubated with 1 µg/µL salmon sperm DNA and 2 µL of 0.01 µM biotin-labeled DNA probes (Appendix A) at room temperature for 30 min. For competition assays, 200-fold of unlabeled DNA probes (Appendix A) were added to the reactions and co-incubated with the biotin-labeled probes. Subsequently, the mixture was incubated at room temperature for 30 min. After electrophoretic separation on a 5% Criterion TBE Polyacrylamide gel (BioRad, Hercules, CA, USA), probes were transferred to a Biodyne B Nylon Membrane (ThermoFisher) followed by UV cross-linking, incubated with avidin-linked HRP, and the signal was measured using the Amersham imager 680 (GE Healthcare, Chicago, IL, USA).

### 2.12. Antibody-Guided Chromatin Tagmentation (ACT)

Histone modifications were done by ACT-qPCR or ACT-seq, largely according to Ref [40]. The pA-Tn5ase protein was isolated from *E. coli* (C3013, New England Biolabs) transformed with plasmid pET15bpATnp (#121137, Addgene). The pA-Tn5 transposome (pA-Tn5ome) was generated by mixing pA-Tn5ase (final concentration either 1.9 µM or 3.3 µM, depending on the pA-Tn5ase preparation) and Tn5ME-A+B load adaptor mix (final concentration 3.3 µM) in complex formation buffer (CB) [40]. The Tn5ME-A+B load adaptor is a mixture of two dsDNA that was generated by pre-annealing the oligonucleotides Tn5MErev with Tn5ME-A and Tn5ME-B, respectively. The pA-Tn5ome-antibody (pA-Tn5ome-ab) complexes were formed by mixing 1 µL pA-Tn5ome with 0.8 µL CB and 0.8 µL antibody solution. The antibodies used in this study were: H3K4me1 (ab8895, Abcam, Cambridge, UK), H3K27ac (ab4729, Abcam), H3k27me3 (CST9733 Cell Signaling, Danvers, MA, USA), yeast histone H2B (M3930, Boster Bio. Tech., Pleasanton, CA, USA), and rabbit IgG (cat #pp64 Millipore, Burlington, MA, USA). Approximately 24,000 nuclei of cell were used for pA-Tn5ome-ab complex binding and tagmentation. For normalization of sequence reads between biological replicates, approximately 3000 permeabilized nuclei of yeast *Saccharomyces cerevisiae* were prepared according to Ref [41], incubated with pA-Tn5ome-ab complex targeting yeast H2B, and were spiked into each mix of cells and pA-Tn5ome-ab complex. Tagmented DNA was purified with a MinElute kit (#28004, Qiagen, Hilden, Germany) and eluted with 20 µL elution buffer (EB). Sequencing libraries were generated under real-time conditions with a LightCycler 480 (Roche, Basel, Switzerland) in 50 µL reaction mixes, which consisted of 20 µL tagmented DNA eluate, 25 µL NEBNext High Fidelity 2X Mix (#M0541, New England Biolabs), 0.5 µL 100X SYBRGreen, 2.5 µL primer Tn5McP1n, and 2.5 µL barcode primer (Appendix A). Reaction conditions were 72 °C, 5 min (gap repair); 98 °C, 30 s (initial melting); 98 °C, 10 s; 63 °C, 10 s; 72 °C, 10 s (cycling). Cycling was stopped when the increase of fluorescence units (FUs) was 5 or higher. Libraries were purified with AMPure XP beads (#A63880, Beckman, Brea, CA, USA) with a bead:DNA ratio of 1.4:1 and 12 µL EB. Quantity and fragment size of the libraries were determined with a Qubit dsDNA HS assay kit (#Q32854, ThermoFisher and a TapeStation 4150 with D1000 High Sensitivity Assay (#5067- 5585, Agilent, Santa Clara, CA, USA), respectively. Samples were further analyzed by qPCR or sequencing as follows.

### 2.13. ACT Followed by Quantification with Real-Time PCR (ACT-qPCR)

Specific enriched regions were quantified using a primaQUANT CYBR green kit (#SL-9902-10mL, Steinbrenner Laborsysteme GmbH, Wiesenbach, Germany) with the indicated primers (Appendix A) using the LightCycler 480 with the following conditions: 95 °C, 15 min; then cycling with 95 °C, 15 s; 55 °C, 30 s; 72 °C, 10 s for 45 cycles. Signals were normalized to IgG signals and reflected the relative expression towards the wild type condition.

### 2.14. ACT Followed by Whole Genome Sequencing (ACT-seq) and Data Analysis

Eight to twelve samples were multiplexed and sequenced on one lane of a NextSeq 550 system (paired-end, 75 bp) with mid-output at the Genome and Proteome Core Facility of the DKFZ. For data processing, Trim Galore v. 0.4.4 (https://www.bioinformatics.babraham.ac.uk/projects/trim_galore/; accessed on 13 December 2020) [42] was applied together with Cutadapt v. 1.14 [43] using the non-default options “--paired”, “--nextera”, “--length_1 35”, and “--length_2 35” to carry out adapter and quality trimming. Trimmed reads were mapped against the Genome Reference Consortium Human Build version 37 by means of Bowtie2 v. 2.2.6 [44] using the “--very-sensitive” flag and a maximum insertion length of 2500 bp. Mappings belonging to the same lane-multiplexed library were combined using SAMtools merge v. 1.5 [45]. Discordant alignments and mappings with a Phred score below 20 were removed using SAMtools view. Adey et al. [46] showed that fragments obtained from tagmentation cannot be smaller than 38 bp. Thus, all alignments corresponding to fragment sizes below that threshold were removed. Read ends were shifted to represent the center of the transposition event as previously described by [47]. Additionally, trimmed reads were aligned against the *Saccharomyces cerevisiae* R64 reference genome followed by post-alignment filtering as described above. To calculate a library-specific scaling factor, we derived the multiplicative inverse of the number of filtered reads mapped to the yeast genome. Coverage tracks were generated by means of the bamCoverage functionality of Deeptools v. 3.1.1 [48] using the non-default parameters “--ignoreForNormalization chrM chrY chrX” and “--effectiveGenomeSize 2652783500” as well as the “--scaleRatio” option to specify the spike-in-based scaling factor. The analysis procedure was implemented as fully containerized workflow using the Common Workflow Language v. 1.0 [49] and is publicly accessible [50]. Signals from each specific region are the sum of each 50 base pairs, and were normalized to signals from the un-irradiated DMSO-treated samples. The regions are: *PROM1*: chr4:16083992-16086985, *SCG3*: chr15:51973407-51973890, *RDH16*: chr12: 57352354-57352795, *DGKA* promoter: chr12: 56324517-56324949 and *DGKA* enhancer: chr12: 56329323- 56330122.

### 2.15. ELISA

The quantification of secreted collagen 1a1 was using the COL1A1 ELISA kit (ab210966, Abcam, Cambridge, UK). In short, conditioned medium from the indicated treatment was collected, diluted (1:500), and further processed according to the manufacturer’s protocol.

### 2.16. Statistics

Statistical significances were determined by either one-tailed Student’s *t*-test or one-way ANOVA, and results with *p*-value < 0.05 were considered as statistically significant. Data were visualized with GraphPad Prism version 5 (GraphPad Software Inc., San Diego, CA, USA). The heatmap graph of DNA methylation was generated using Morpheus (https://software.broadinstitute.org/morpheus; accessed on 13 July 2020).

## 3. Results

### 3.1. DNA Demethylation of the DGKA Enhancer Region Results in Increased DGKA Expression after Irradiation

To verify whether the DNA methylation pattern at the *DGKA* enhancer region affects EGR1-binding (Figure 1A), we performed an in vitro electrophoretic mobility shift assay (EMSA) using two oligonucleotide probes (E1 or E2 probe), each covering one of the wild-type (WT) sequences of the two EGR1-binding sites (EGR1_1 and EGR1_2). A band shift was observed when EGR1-overexpressing cell lysates were co-incubated with the unmethylated E1 or E2 probe (Figure 1B, lane 1 and lane 5). The intensity of this band shift was strongly reduced by 90% after incubation with the methylated E1 probe (E1me), but not altered when incubated with the methylated E2 (E2me) probe (Figure 1B, lane 1 versus 3 and lane 5 versus 7). This indicated that DNA methylation of the first EGR1-binding site impedes binding of the transcription factor.

Next, we demethylated the *DGKA* enhancer by CRISPR/dCas9-based epigenomic editing in human embryonic kidney HEK293T cells (Figure 1A), where this region is highly methylated and endogenous *DGKA* expression is low. The nuclease-dead Cas9 (dCas9) was fused with the tripartite activator, VP64-p65-Rta (dCas9-VPR) [51] and a methylcytosine dioxygenase TET3 (named here dCas9-VPR-TET3) and transfected into cells by a cumate-inducible vector system. dCas9-VPR-TET3 expression was already detectable (1.3-fold increase) after transfection, indicating leakiness of the vector system. However, cumate treatment strongly increased expression of dCas9 (4.6-fold; Appendix A). Methylation was measured by MassArray analysis covering eight CpG units at this region with mean β-values for DNA methylation from 0 to 1. Compared to un-transfected cells, DNA methylation of the *DGKA* enhancer was reduced from a β-value of 0.9 to 0.8 in transfected cells without cumate treatment, and further down to 0.7 in cumate-induced transfected cells (Figure 1C). Site-specific DNA methylation at all eight CpG units was reduced (Figure 1D). Endogenous *DGKA* expression in the demethylated cells was slightly higher than in un-irradiated HEK293T WT cells (1.4-fold increase, *p* = 0.126), and *DGKA* induction was further increased after irradiation (see the fold-increase for dCas9-VPR-TET3 from 1.4 (−IR) to 3.1 (+IR), *p* = 0.017; and compared the fold-increase of 1.7 (WT +IR) to 3.1 (dCas9-VPR-TET3 +IR) after irradiation, *p* = 0.002; Figure 1E), confirming that DNA methylation at the *DGKA* enhancer modulates radiation-induced *DGKA* expression. *EGR1* expression was similar in HEK293T and dCas9-VPR-TET3-expressing cells with or without cumate treatment (Appendix A). Because pro-fibrotic markers are not expressed in HEK293 cells (compared to foreskin and brain fibroblasts or skin keratinocytes; Appendix A, [52]), an increase of these markers was not detectable in demethylated HEK293T cells.

Finally, we investigated whether the reduced DNA methylation and VPR recruitment are coupled to changes in histone modifications, and used antibody-guided chromatin tagmentation (ACT)-qPCR to analyze the active chromatin marks histone H3 lysine 4 mono-methylation (H3K4me1) and H3 lysine 27 acetylation (H3K27ac), as well as the repressive mark H3 lysine 27 tri-methylation (H3K27me3) at the *DGKA* enhancer region. H3K4me1 was enriched more than 5-fold at both EGR1-binding sites in the demethylated cells compared to WT cells (*p* = 0.018 and 0.001), but not at the upstream *DGKA* promoter or downstream control sites (Figure 1F). H3K27ac enrichment occurred at the second EGR1-binding site in demethylated cells (3-fold enrichment, *p* = 0.034; Figure 1G). H3K27me3 was enriched in demethylated cells, but this enrichment occurred nonspecifically at all tested sites, including the upstream promoter and downstream control sites (Figure 1H). These results indicate that demethylation of the EGR1-binding sites at the *DGKA* enhancer region and VPR recruitment are associated with active chromatin marks, favoring increased *DGKA* transcription after irradiation, potentially also by EGR1-binding.

### 3.2. Loss of EGR1-Binding at the DGKA Enhancer Suppresses Induction of DGKA and COL1A1 after Irradiation

We edited the two EGR1-binding sites at the *DGKA* enhancer by CRISPR/Cas9 technology to study the function of EGR1-binding in radiation-induced *DGKA* expression and its consequences for pro-fibrotic marker expression in human foreskin BJ fibroblasts. We selected one sgRNA directed against each EGR1-binding site and generated two cell lines with edited EGR1-binding sites, called sgE1 and sgE2 mutant cells (Figure 2A). Sanger sequencing verified a one-base insertion in sgE1, which was located 14 bp 5′ of the first EGR1-binding site. Targeting the EGR1-binding site directly was not possible because of the required proto-spacer adjacent motif (PAM) for the sgRNA. The sgE2 mutant consisted of a five-base pair deletion in the second EGR1-binding site (Figure 2A), which interrupted the TF-binding motif. Control cells (sgC) were infected with lentivirus carrying a non-targeting sgRNA (Lenti-CRISPR-sgRNA-Gal sgRNA) and retained the WT sequence.

We demonstrated by EMSA that the binding affinity of EGR1 to the *DGKA* enhancer was altered after gene editing. EGR1 from overexpressing cell lysates was bound to the biotinylated E1 probe, which represented the first EGR1-binding site, leading to a DNA-protein band shift (Figure 2B, lane 2). The intensity of the upper band was reduced 0.5-fold in the presence of E1WT (lane 3) but not in the presence of E1mut and BD1 as competitors (lanes 4 and 5). E1mut contains the same insertion as sgE1 cells and BD1 contains a GGCG to ACTA mutation in the first EGR1-binding site. A band shift pattern was detected, including a faint double-band when the E2WT probe and EGR1-overexpressing cell lysates were co-incubated (lane 7). The faint double-band was lost in the presence of the E2WT competitor, but not in the presence of the E2mut competitor with the same deletion sequence as in sgE2 cells (lane 8 and 9). Therefore, these results suggest that the EGR1-binding affinity at the two predicted EGR1-binding sites was abolished in each of the two genome-edited BJ cell lines. Of note, DNA methylation at the *DGKA* enhancer region was not significantly changed by genome editing (Appendix A).

Next, we irradiated the two mutant cell lines sgE1 and sgE2 to analyze induction of *DGKA* and pro-fibrotic marker expression. Irradiation significantly increased *DGKA* levels in sgC cells (1.7-fold change after 2 Gy and 2.9-fold change after 6 Gy irradiation), but not in sgE1 and sgE2 mutant cells (Figure 2C). Expression of the pro-fibrotic marker collagen 1 alpha 1 (*COL1A1*) was significantly increased after irradiation in sgC cells (almost 3-fold change in cells irradiated with 2 and 6 Gy compared to un-irradiated cells), but not in sgE1 and sgE2 mutant cells (Figure 2D). After irradiation, COL1A1 pro-peptide secretion into the culture medium was significantly increased in sgC (1.4-fold change, *p* = 0.028) but not in sgE1 and E2 cells (Figure 2E). Other pro-fibrotic markers including *COL3A1*, alpha smooth muscle actin (*ACTA2*), and fibronectin 1 (*FN1*) were not significantly induced after irradiation in sgC and mutated cells (Figure 2F and Appendix A). In summary, our data show that EGR1-binding at the *DGKA* enhancer controls *DGKA* induction and the production of pro-fibrotic COL1A1 after irradiation.

### 3.3. Genome-Wide Effects Induced by Loss of EGR1-Binding at the DGKA Enhancer

A transcriptome analysis was performed with edited BJ cells to identify potential genome-wide consequences caused by the loss of EGR1-binding at the *DGKA* enhancer. A multi-factor analysis comparing the isogenic sgC and sgE1 cell lines allowed the identification of both genome editing and IR effects (Figure 3A). RNA sequencing was done 48 h after irradiation with 6 Gy using four biological replicates for each condition. A multidimensional scaling plot revealed a strong separation of the samples by radiation (dimension 1, Figure 3B and Appendix A), but no separation of sgC and sgE1 cells based on the first two dimensions, either without or with irradiation. This suggests that the effect of IR was greater than that of the loss of EGR1-binding.

In more detail, the loss of EGR1-binding at the *DGKA* enhancer caused only 22 upregulated and 12 downregulated differentially expressed genes (DEGs) in un-irradiated cells (Figure 3C, Appendix A). A Gene Ontology (GO) analysis showed significant enrichment of these DEGs in 4 pathways related to platelet activation, mitochondrial translation, and insulin response (Appendix A). The comparison of irradiated with un-irradiated cells revealed 967 DEGs (247 upregulated and 720 downregulated) in sgC cells (Appendix A) and 2227 DEGs (929 upregulated and 1298 downregulated) in sgE1 cells (Appendix A). After irradiation, most of the top 20 GO pathways (Appendix A) were related to DNA replication and repair pathways, with nine pathways being present in both sgC and sgE1 cells, thus highlighting a strong effect of IR. After adjustment for irradiation, multi-factorial analysis of the effect of EGR1-binding at the *DGKA* enhancer revealed 272 DEGs (144 were upregulated and 128 downregulated) in sgE1 cells (Figure 3D, Appendix A). A strong GO enrichment was observed for fibroblast activation pathways (highlighted in red) including negative regulation of endopeptidase activity, collagen fibril organization, and extracellular matrix organization (Figure 3E). Several dysregulated genes show up in multiple pathways (Appendix A). Many of them, including collagen genes as well as *COMP*, *PDGFA, ADRA2A, FOXC2,* and *FLRT2* have been reported to be related to fibrosis. Taken together, these results support the hypothesis that loss of EGR1-binding at the *DGKA* enhancer by genomic editing leads to reduced radiation-induced *DGKA* expression. This reduction affects multiple potential biological functions and pathways related to fibroblast activation, which becomes visible only after irradiation of fibroblasts.

### 3.4. Treatment with EZH1/2 Inhibitors Did Not Affect Radiation-Induced DGKA Expression and Suppressed the Induction of a Pro-Fibrotic Response

Our results on (epi)genomic editing of the *DGKA* enhancer suggest an *EGR1-DGKA-COL1A1/3A1* gene induction axis to regulate pro-fibrotic gene expression in fibroblasts. To investigate how to interrupt this axis, we used pharmaceutical drugs to modulate the epigenetic pattern at the *DGKA* enhancer region. Two EZH1/2 inhibitors, GSK126 and EPZ6438, were used in BJ cells at concentrations yielding more than 75% cell viability (5 μM GSK126 and 50 μM EPZ6438). Irradiation with 6 Gy did not further reduce the viability of EZH1/2 inhibitor-treated cells (Appendix A). Even though nuclear H3K27me3 levels were not significantly affected by inhibitor treatments (Figure 4A), enrichment of this mark was reduced by both drugs at genes not expressed in fibroblasts such as *PROM1* (a cancer stem cell marker), *SCG3* (expressed in brain tissue), and *RDH16* (expressed in liver tissue) (Figure 4B), indicating efficient inhibition of the histone methylases EZH1/2. However, H3K27me3 enrichment was not detected at the *DGKA* enhancer region in BJ cells (Appendix A). This could explain why baseline expression of *DGKA* was not changed by the drug treatment in BJ cells. Consequently, *DGKA* expression was induced after irradiation and this induction was not significantly altered by the drugs (*p* = 0.142 for GSK126 and *p* = 0.153 for EPZ6438; Figure 4C). *EGR1* expression was not affected by either drug treatment (Appendix A).

Radiation-induced expression of *COL1A1* was induced (1.7-fold induction, *p* = 0.019) compared to DMSO-treated, un-irradiated cells, but this induction was reduced by GSK126 and by EPZ6438 (compare the fold change of 1.8 to 1.4, *p* = 0.039 for GSK126 and to 0.4, *p* = 0.007 for EPZ6438; Figure 4D). For *COL3A1*, radiation-induced expression was induced to 1.4-fold (*p* = 0.059) compared to un-irradiated, DMSO-treated cells. This induction was not altered by GSK126 (compare the fold change of 1.4 to 1.5, *p* = 0.074), but blocked by EPZ6438 (compare the fold change of 1.4 to 0.5, *p* = 0.035; Figure 4E). Radiation increased COL1A1 secretion from 27 to 43 ng/mL (*p* = 0.031) in DMSO-treated cells (Figure 4F). Radiation-induced secretion was reduced to 33 ng/mL by GSK126 (*p* = 0.148) and to 2 ng/mL by EPZ6438 (*p* = 0.011). The results from H3K27me3 levels and radiation-induced gene expression indicate that *DGKA* expression is differently regulated than that of *COL1A1* and *COL3A1*.

To further investigate the role of EZH1/2 in the induction of *DGKA* and pro-fibrotic markers after irradiation, we reduced EZH1 and EZH2 levels by siRNA in BJ cells. Compared to the scrambled siRNA (siScr)-treated cells, the expression of *EZH1* and *EZH2* was reduced by 70 and 85% after separate knockdown, and by 60 and 70% after double knockdown (Appendix A and Figure 4G). Neither single nor double knockdown decreased radiation-induced expression of *DGKA* or the pro-fibrotic markers as compared to control cells (Appendix A, and Figure 4H), indicating that EZH1/2 may not regulate *DGKA* and collagen expression.

Next, we treated mutant sgE1 cells with GSK126 and EZP6438. As already observed (Figure 2C–E), radiation-induced *DGKA* and *COL1A1* expression was abolished after enhancer editing in DMSO-treated cells (compare the fold increase after irradiation of 2.8 to 1.3 for *DGKA*, Appendix A; 3.7 to 1.4 for *COL1A1*, Appendix A) and this reduction was not further changed when treated with GSK126 (compare the fold change of 2.8 to 0.9 for *DGKA*, Appendix A; 2.4 to 1.1 for *COL1A1*, Appendix A) and EPZ6438 (compare the fold change of 1.7 to 1.3 for *DGKA*, Appendix A; 0.6 to 0.5 for *COL1A1*, Appendix A). As observed for the EZH1/2 single and double knockdown (Appendix A and Figure 4G), the radiation-induced *COL3A1* expression was, however, boosted in sgE1 cells when treated with GSK126 (compare the fold change of 0.4 to 2.6) and EZP6438 (compare the fold change of 0.5 to 2.1) (Appendix A). Altogether, the results of H3K27me3 and radiation-induced *DGKA* induction indicated that epigenetic regulation of the *DGKA* enhancer by EZH1 and EZH2 is not involved in the radiation-induced *EGR1-DGKA-COL1A1/3A1* axis in BJ fibroblasts. Inhibitors, however, can block the expression of pro-fibrotic collagens after irradiation suggesting further pro-fibrotic regulatory mechanisms.

### 3.5. Bromodomain Inhibitors Attenuate Radiation-Induced DGKA and Pro-Fibrotic Marker Expression in BJ Cells

Active chromatin is characterized by the H3K27ac mark, which facilitates transcription. To study the role of this mark at the *DGKA* enhancer, we used the epigenetic drug JQ1, which inhibits BET-containing readers of H3K27ac [24], and CBP30, which specifically inhibits the histone acetyltransferase CBP/p300 [53]. We used a concentration of 5 µM for JQ1, which was previously used in normal human fibroblasts [25]. This concentration yielded more than 50% viability, while 10 µM of CPB30 resulted in more than 80% viability (Appendix A). Irradiation did not further suppress cell viability for either drug treatment (Appendix A). JQ1 and CBP30 treatment did not significantly change the nuclear amount of H3K27ac in the un-irradiated cells, but reduced it after irradiation (reduced to 0.7- and 0.6-fold for JQ1 and CBP30) compared to DMSO-, un-irradiated cells, demonstrating the successful uptake of inhibitors (Figure 5A). Enrichment of H3K27ac at the *DGKA* promoter and enhancer was not significantly suppressed by CBP30 treatment and irradiation (Figure 5B). We observed, however, a large variability after both treatments in the biological replicates, which might have masked potential effects.

Even though expression of the radiation-induced transcription factor *EGR1* was significantly increased when the cells were treated with the two bromodomain inhibitors (Appendix A), radiation-induced *DGKA* expression was blocked by both drugs (Figure 5C). Endogenous expression of the pro-fibrotic markers *COL1A1* and *COL3A1*, was nearly completely abolished by JQ1 treatment compared to DMSO-treated cells (compare the fold change of 1.0 to 0.2 for both *COL1A1* and *COL3A1*) and it was not induced after irradiation (Figure 5D,E). For CBP30-treated cells, endogenous *COL1A1* was not significantly altered in the un-irradiated cells compared to DMSO-treated cells (compare the fold change of 1.0 to 0.73, *p* = 0.051), and radiation-induced expression was suppressed (compare the fold change of 2.4 to 1.0, *p* = 0.004 Figure 5D). Endogenous *COL3A1* was reduced in CBP30-treated cells compared to DMSO-treated cells (compare the fold change of 1.0 to 0.4, *p* = 0.001), and it was not induced after irradiation (Figure 5E). Baseline COL1A1 secretion was reduced by 90% through JQ1 and not further changed after irradiation. For CBP30, baseline COL1A1 was unaltered with or without irradiation (Figure 5F). These results support that both JQ1 and CBP30 affect H3K27ac, and can attenuate the induction of *DGKA* and pro-fibrotic marker expression by irradiation, resulting in reduced fibroblast activation.

BRD2 and BRD4 were shown to be JQ1 targets involved in bleomycin-induced *DGKA* expression [25], and we therefore assumed that this might be similar for the radiation-induced effects. To verify the specific inhibition of CBP and p300 by CBP30, we silenced both proteins by siRNA in BJ cells. Compared to siScr-treated cells, residual expression of *CBP* and *p300* was 26 and 36% when the cells were transfected with either siCBP or sip300 (Appendix A), and was about 40 and 60% when both proteins were silenced together (Figure 5G). Only the CBP and p300 double knockdown, but not the single knockdowns, suppressed the induction of *DGKA* and pro-fibrotic markers after irradiation treatment (Figure 5H and Appendix A). Remarkably, we again observed a strong reduction of about 80% of *COL1A1* expression in un-irradiated CBP and p300 double knockdown cells.

Next, we investigated the BET inhibitors in cells harboring a mutation site at the *DGKA* enhancer in order to verify whether there is a DGKA-independent inhibitor effect. There was no radiation-induced *DGKA* expression in either DMSO- or bromodomain inhibitor-treated sgE1 cells (Appendix A). Hence, the radiation-induced expression of *COL1A1* and *COL3A1* was also blocked in sgE1 cells (Appendix A). Taken together, these results show that BET protein-mediated epigenetic activity is required for the induction of *DGKA* and pro-fibrotic markers by irradiation.

### 3.6. Effects of Epigenetic Inhibitors in Human Dermal Fibroblasts with Low or High DNA Methylation at the DGKA Enhancer

In order to verify whether the inhibitory drug effects depend on the variation in methylation status of this genomic region, we used primary normal human dermal fibroblasts (NHDFs) from 7 donors that differ in the DNA methylation level of the *DGKA* enhancer. Fibroblasts L1 to L4 and BJ cells showed low methylation values with β < 0.6 whereas fibroblasts H1 to H3 showed high values with β > 0.7 (Figure 6A). Site-specific DNA methylation at CpG_6, which is located within the first EGR1-binding site was low in L1 to L4 and BJ cells, and high in H1 and H2 (Figure 6B). Therefore, we classified the methylation status of L1 to L4 and BJ cells as “low” or “hypomethylated” and that of H1 to H3 as “high” or “hypermethylated”. In the DMSO control groups, mean induction of *DGKA* expression was up 1.7-fold (*p* = 0.06) after irradiation in the cells with low methylation, while there was no induction in those with high methylation (Figure 6C). Radiation-induced *COL1A1* and *COL3A1* expression was significantly increased in the low methylation group (2.5-fold change, *p* = 0.045 for *COL1A1*; 2.4-fold change, *p* = 0.025 for *COL3A1*), whereas induction was only about 1.3- and 1.2-fold in the high methylation group (Appendix A). These results further confirm that the DNA methylation status of the *DGKA* enhancer affects radiation-induced pro-fibrotic marker expression, also in primary fibroblasts.

Next, we treated primary fibroblasts with low and high DNA methylation with EZH2 inhibitors. Endogenous *DGKA* expression was reduced after drug treatment, especially in the group of fibroblasts with high methylation after treatment with EPZ6438 (0.27-fold change, *p* = 0.01). Radiation-induced *DGKA* expression was not significantly altered by either drug in low and high methylated NHDFs (Figure 6C). Levels of endogenous *COL1A1* and *COL3A1* were slightly increased in low and high methylated NHDFs when treated with GSK126 and EPZ6438, respectively (Appendix A). Significant induction of *COL1A1* and *COL3A1* after irradiation was not observed in EZH2 inhibitor-treated NHDFs with low and high methylation status. Taken together, our results in NHDFs show that the EZH1/2 inhibitors did not affect radiation-induced *DGKA* expression and fibroblast activation, depending on the DNA methylation status.

To further verify whether bromodomain inhibitors can attenuate radiation-induced fibroblast activation, we treated fibroblasts exhibiting low or high DNA methylation with JQ1 and CBP30. JQ1 suppressed both endogenous and radiation-induced *DGKA* expression in both fibroblast groups down to 50%. CBP30 mainly abolished this induction in low methylated NHDFs (compare fold change from 1.2 to 0.9, *p* = 0.048; Figure 6D). JQ1 further abolished both endogenous and radiation-induced *COL1A1* and *COL3A1* expression in low and high methylated NHDFs (Appendix A). CBP30 significantly suppressed the radiation-induced *COL1A1* expression in both NHDF groups (compare the fold change of 2.4 to 1.1 for low methylated NHDFs and 1.3 to 0.4 for high methylated NHDFs, Appendix A). Suppression of *COL3A1* induction by CBP30 was only observed in low methylated NHDFs (compare the fold change of 2.1 to 0.6 for low methylated NHDFs and 1.2 to 0.6 for high methylated NHDFs, Appendix A) as there was no significant induction of *COL3A1* in high methylated cells. Overall, these results show that bromodomain inhibitors may be potential drugs to prevent radiation-induced fibroblast activation mediated by DGKA, especially in patients with DNA hypomethylation at the *DGKA* enhancer.

## 4. Discussion

Radiation-induced expression of DGKA regulated by the differentially methylated *DGKA* enhancer region was previously reported to be associated with higher fibrosis risk in fibroblasts of breast cancer patients who received radiotherapy [12]. In the present study, we hypothesized a pro-fibrotic *EGR1*-*DGKA-COL1A1/3A1* gene induction axis after irradiation in fibroblasts. We therefore modulated the *DGKA* induction by epigenome and genome editing of the enhancer, as well as by drugs targeting epigenetic regulators (Appendix A). The aim of this modulation was to attenuate the radiation-induced pro-fibrotic response and, as a long-term goal, to reduce fibrosis development.

DNA demethylation of the *DGKA* enhancer by the dCas9-VPR-TET3 system in HEK293T cells was accompanied by enrichment of the activating histone marks H3K4me1 and H3K27ac [54,55] and enhanced *DGKA* expression after irradiation. Strong synergistic effects of the VPR transactivation domain or similar activation domains and the TET enzyme have been observed [56,57], therefore the transcriptional activation and deposition of activating histone markers might not only be caused by demethylation and EGR1 binding but partly also due to VPR-TET3 recruitment at the demethylated *DGKA* enhancer. In addition, an enrichment of the repressive histone mark H3K27me3 was observed after cumate treatment. This enrichment was not specific for the demethylated region and not as strong as the enrichment for H3K4me1. The co-existence of H3K27me3 and H3K4me1 is described for primed enhancers, which play a role during development and differentiation [58]. The presence of both histone marks might indicate that the reactivation of the *DGKA* element by epigenetic editing might involve a transition state similar to primed enhancers. Finally, H3K27me3 inactivation might be overcompensated by the two active marks, H3K4me1 and H3K27ac, allowing radiation-induced *DGKA* expression. Besides, chromatin accessibility in fibrosis was described for multiple genomic regions including the *DGKA* locus in fibroblasts from pulmonary fibrosis patients after activation of the transcription factor *JUN* [59], suggesting that the chromatin configuration has an essential role not only in radiation-induced fibrosis, but also in other types of fibrosis. Our results in NHDFs with high and low methylation support this observation.

Radiation-induced *DGKA* expression was mediated by the stress-inducible transcription factor EGR1 [12]. Inactivation of the EGR1-binding sites by genomic editing in fibroblasts abolished *DGKA* induction and expression and secretion of COL1A1, underlining the regulatory role of the *DGKA* enhancer region for pro-fibrotic processes. Even the one base pair insertion upstream of the EGR1-binding site, as in sgE1 cells, affected *DGKA* induction. One explanation might be that the mutation created a new binding motif for another transcription factor. Binding of the additional TF might impede EGR1-binding and the related transcriptional effects. This hypothesis is supported by a study of Sun and colleagues on brain development, which indicated that EGR1, although it can bind to methylated DNA, may prevent the binding of other TFs adjacent to the EGR1-binding site in this way impeding their function [60].

Comparing the transcriptomes of edited and control fibroblasts without irradiation showed only minor baseline expression differences. This confirms the high specificity of gene editing leading to isogenic cell lines, although we cannot exclude expression differences due to selection process of edited cells. Expression changes after irradiation were substantial and mainly related to DNA repair response, replication arrest, and apoptosis, both in control and genome-edited cells. Further interesting differentially expressed genes were identified after editing and adjustment for irradiation. These genes were enriched in specific GO pathways, which were related to extracellular matrix metabolism and other processes involved in fibrosis such as collagen fibril organization, skin development, extracellular matrix organization, cell adhesion, cellular response to endogenous stimulus, and cellular response to growth factor stimulus [61,62,63,64]. Dysregulation of endopeptidase activity might disturb the proper maturation of collagen fibrils as suggested in myocardial fibrosis [65]. The most prominent genes include *COL1A1, COL3A1*, *COMP, PDGF, ADRA2A, FOXC2,* and *FLRT2*. Both collagens were already shown to be inducible by *DGKA*. COMP is a glycoprotein involved in collagen secretion and fibrillogenesis [66] and upregulated in fibroblasts after UVA irradiation [67]. The protein is detected in serum and skin biopsies from systemic sclerosis patients with fibrotic skin lesions [68,69]. PDGF is a key pro-fibrotic growth factor that plays an important role in the development of fibrotic diseases including IR-induced fibrosis [70,71]. ADRA2A encodes a subtype of the adrenergic receptor family and is reported to be expressed in hepatic stellate cells and non-tumor fibrotic liver tissue [72]. FOXC2 is a transcription factor that can trigger epithelial-mesenchymal transition during organ repair [73,74]. FLRT2 can interact with fibronectin and promotes cell proliferation during chondrogenesis [75,76]. Taken together, these differentially expressed genes and their functions in fibrosis support our observation that loss of radiation-induced *DGKA* upregulation attenuates genes, which are involved in the activation of fibroblasts, extracellular matrix production, and cell–cell interaction. Future and more detailed investigations, which were beyond the scope of our current investigation, are needed to elucidate the potential of these dysregulated genes as targets to treat fibrosis.

Targeting a pro-fibrotic epigenetic pattern by small molecules, which affect epigenetic players, offers an attractive therapeutic approach to attenuate fibrosis risk in radiotherapy patients. First, we targeted the activity of EZH1/2 methyltransferases, which regulate gene expression as part of the PRC2 complex by inducing the repressive mark H3K27me3. This approach was based on studies that point out that (i) EZH2 expression is positively correlated with TGFβ1, a main player in fibrosis [20,77], and (ii) loss of EZH2 activity by genetic or pharmacologic blockade can reverse fibrosis progression in various tissues [20,21,22,77,78]. However, the published evidence highlighting the contribution of EZH2 to fibrosis is not unequivocal. A study by Grindheim and colleagues indicated that loss of *Ezh1* and *Ezh2* in mouse hepatocytes dysregulated postnatal hepatic maturation, ultimately leading to chronic liver damage and fibrosis [79]. Also, in a diabetic nephropathy model, TGFβ-induced pro-fibrotic genes such as connective tissue growth factor (*Ctgf*) and serpin family E member 1 (*Serpine1*) are increased when *Ezh2* is depleted by siRNA in renal mesangial cells [80].

We applied the inhibitors GSK126 and EPZ26438 to human dermal fibroblasts at concentrations able to inhibit both EZH1 and EZH2 and to reduce H3K27me3 at specific genomic sites. Both inhibition and silencing of EZH1 and EZH2 did not attenuate radiation-induced *DGKA* expression in BJ cells and in NHDFs with low DNA methylation. One reason might be that H3K27me3 was not detected at the promoter and enhancer region of *DGKA* in BJ cells. In contrast, in inhibitor-treated cells, *COL1A1* and *COL3A1* were not induced by irradiation corresponding to literature findings regarding other types of fibrosis [20,21,22]. Therefore, our results indicate that attenuation of pro-fibrotic markers by EZH1/2 inhibition is not mediated by DGKA. Further mechanisms like the TGFβ-mediated signaling pathways [20,77] and the ribonucleoprotein complex MiCEE function (Mirlet7d-C1D-EXOSC10-EZH2) [81] might be involved. Remarkably, *COL3A1* was induced when silencing *EZH1* and *EZH2* separately or together. This underlines that pharmacological enzyme inhibition shows different effects than reduction of participating proteins in the PRC2 complex formation, an observation that might be caused by the described dual role of EZH2 [17]. Thus, targeting EZH1/2 will not affect fibroblast activation after irradiation via the DGKA-mediated pro-fibrotic axis, and might, thus, not be beneficial as a treatment for radiation-induced fibrosis.

Second, we used bromodomain inhibitors to block recognition and reading of *DGKA* enhancer-associated H3K27ac marks [82]. In our experiments with human fibroblasts, JQ1 treatment inhibited the induction of *DGKA* and pro-fibrotic markers after irradiation. Especially *COL1A1* expression was nearly completely reduced. These data confirm similar results in fibroblasts treated with bleomycin, a radiomimetic drug [25]. JQ1 is a pan-BET inhibitor, which can target the two tandem bromodomains of BET proteins such as BRD2, BRD3, and BRD4, which are involved in transcriptional co-activation [24]. The drug has been shown to attenuate tumor-associated fibrosis in the pancreas [83] and irradiation-induced lung fibrosis [84]. However, JQ1 exerted severe cellular toxicity not only in our cell system but also in clinical applications [85], thus impeding its successful use in patients.

Third, as H3K27ac was reduced after JQ1 treatment, we also investigated the effect of CBP30, a highly selective inhibitor of the HAT protein complex CBP/p300, which induces the H3K27ac mark [86]. The compound has been suggested to reduce fibrosis because it either directly or together with an inhibitor of the main collagen receptor discoidin domain receptor 1 (DDR1) attenuated lung inflammation and fibroblast activation in idiopathic pulmonary fibrosis [81,87]. In addition, a transcriptome analysis found that multiple pro-fibrotic pathways were dysregulated in CBP30-treated myofibroblasts derived from patients with Dupuytren’s disease [88]. As to be expected, CBP30 reduced the overall H3K27ac level and the radiation-induced expression of *DGKA* and collagens in BJ fibroblasts. The high variability of the H3K27ac mark after irradiation at the *DGKA* enhancer might be in contrast to the overall reduction of the mark after CBP30 treatment, but might be indicative of cell plasticity and a range of transition states in the treated fibroblasts. Our result was further confirmed by silencing the CPB/p300 complex via siRNAs, but not with the pan-HAT inhibitor PU139, which did not inhibit *DGKA* induction in bleomycin-treated fibroblasts and which shows only limited activity for p300 [25].

Thus, our data show that inhibition of the reading or writing of the transcription-activating histone mark H3K27ac may reduce the induction of pro-fibrotic processes after irradiation. With regard to the differential DNA methylation at the *DGKA* enhancer, it is obvious that H3K27ac in the chromatin pattern requires unmethylated DNA, which was found in the fibroblasts of breast cancer patients with increased fibrosis risk [12]. Therefore, we validated our results in normal human fibroblasts with high and low DNA methylation at the *DGKA* enhancer. In fact, *DGKA* was induced by irradiation only in fibroblasts with low DNA methylation and this induction was inhibited by the two bromodomain inhibitors (JQ1 and CBP30) applied in this study. This different inhibitory effect was further found as different *COL1A1* and *COL3A1* inhibition in low and high methylated NHDFs, although the drugs show differences in target enzyme and cell toxicity. Of course, our results from two-dimensional fibroblast cultures are limited, especially with regard to the intra-cellular and tissue-specific processes occurring during wound-healing and fibrosis development. Therefore, drug effects have to be further scrutinized in additional pre-clinical models before they can be explored for their translational potential. Models for radiation-induced fibrosis exist, but mainly focus on mice [89]. However, the differentially methylated region of the *DGKA* enhancer is not highly conserved and exists only in primates. Animal models targeting the *DGKA* enhancer could be created by genetic engineering and could help to bridge the gap between bench and bedside for these drugs.

This also concerns the toxicity of these drugs in fibroblasts, which is rather high in the case of JQ1. Most recently, more selective BET inhibitors have been developed, which specifically target the two tandem bromodomains (BD1 and BD2) of BET family proteins [90]. The more specific BET-BD2 inhibitor predominantly affected inflammation or immune-related cellular responses leading e.g., to liver fibrosis. This drug might also be promising to inhibit radiation-induced cellular reactions, which include inflammatory and immune reactions. Thus, the use of specific small molecule inhibitors with low toxicity might attenuate radiation-induced fibrotic reactions.

## 5. Conclusions

We modulated radiation-induced *DGKA* expression by epigenomic and genomic editing of the *DGKA* enhancer and administering epigenetic drugs. Demethylation using a CRISPR/dCas9-coupled VPR-TET3 fusion protein construct resulted in an increase of enhancer-related histone marks and caused *DGKA* induction after radiation in HEK293T cells. Editing the binding-sites of the stress-inducible transcription factor EGR1 at the *DGKA* enhancer by a targeted CRISPR/Cas9 approach in BJ fibroblasts decreased radiation-induced *DGKA* and pro-fibrotic marker expression and caused dysregulation of multiple fibrosis-related pathways. EZH2 inhibitors (GSK126, EPZ6438) had no effect on radiation-induced *DGKA* transcription but reduced induction of COL1A1 in BJ cells. Bromodomain inhibitors (CBP30, JQ1) targeting the histone acetyltransferase CBP/p300 or acetylation sensitive BET protein, like BRD4 suppressed the radiation-induced *DGKA* and pro-fibrotic marker expression. Drug effects were confirmed in dermal fibroblasts with low DNA methylation at the *DGKA* enhancer derived from female donors. Our results reveal how epigenetic regulation of the *DGKA* enhancer region contributes to pro-fibrotic reactions. Based on our data, clinical application of bromodomain inhibitors will open promising ways to epigenetically modulate *DGKA* expression as a novel option for a personalized treatment to attenuate long-term fibrosis risk after radiotherapy.

## Figures and Tables

**Figure 1 cancers-13-02455-f001:**
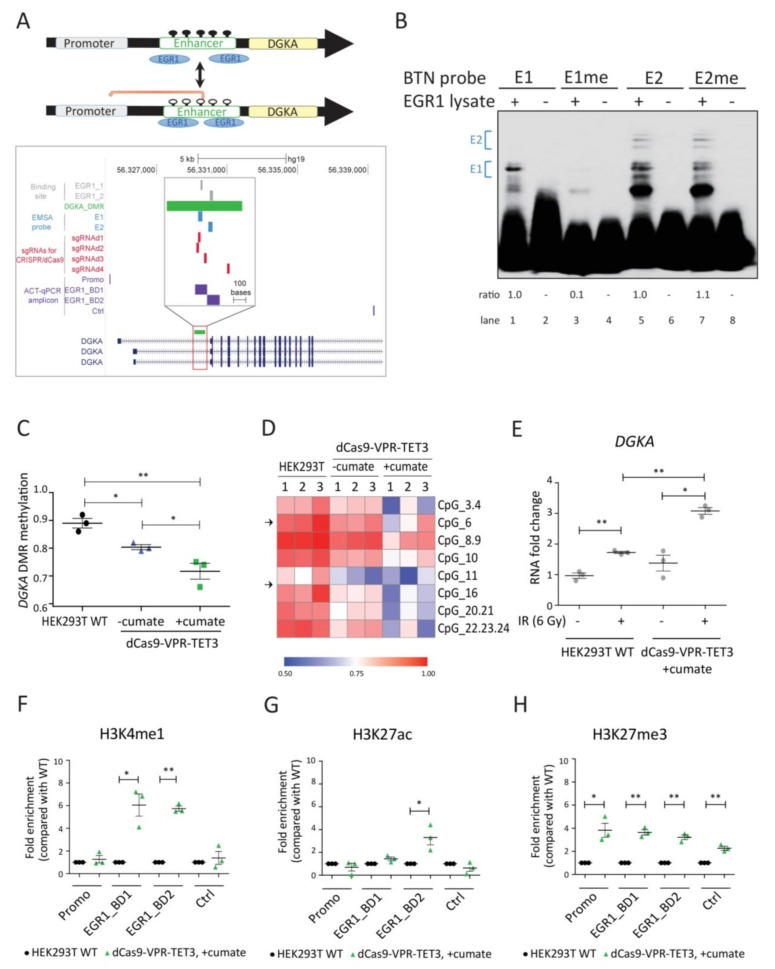
DNA demethylation of the *DGKA* enhancer region results in increased *DGKA* expression after irradiation. (**A**) Upper panel: Schematic presentation of the epigenetic regulation at the *DGKA* locus. Black dots represent methylated CpG sites and white circles unmethylated CpG sites. Lower panel: Map of the interrogated *DGKA* enhancer region indicating the two EGR1-binding sites (EGR1_1 and EGR1_2, gray), the differentially methylated region (DGKA_DMR) and the amplicon for the EpiTYPER assay (green), regions for EMSA probes (E1 and E2, blue), four sgRNA-targeting sites for CRISPR/dCas9-VPR-TET3 gene editing (red), and amplicons for antibody-guided chromatin tagmentation (ACT)-qPCR (Promo, EGR1-BD1 and EGR1-BD2, Ctrl, purple) and *DGKA* transcripts (navy). Amplicons for ACT-qPCR cover the promoter region (Promo), the two EGR1-binding regions (EGR1_BD1 and EGR1_BD2) and the 3‘ downstream control region (Ctrl). (**B**) DNA-binding activity of EGR1 at the *DGKA* enhancer measured by EMSA. E1/E1me and E2/E2me denote the biotin-labeled probes (BTN probes) that contain unmethylated/methylated oligonucleotides, which cover the first or second EGR1-binding site. Ratios indicated are based on the E1 or E2 band shifts highlighted with blue brackets. (**C**) Methylation average across *DGKA* DMR-associated CpGs in HEK293T WT cells as well as dCas9-VPR-TET3 expressing cells with or without cumate (300 µg/mL) treatment. Methylation was measured as β-values by EpiTYPER technology. (**D**) Heatmap of DNA methylation of all CpG sites measured as described in (**C**). For each cell type, three replicates (1,2,3) are shown. Arrows indicate the location of the EGR1-binding motifs. The first EGR1-binding site (EGR1_1) covers CpG_6, and the second binding site (EGR1_2) covers CpG_14, which is located between CpG_11 and CpG_16 and not detected by EpiTYPER. (**E**) Relative mRNA expression of *DGKA* in HEK293T and dCas9-VPR-TET3-expressing cells treated with cumate (300 µg/mL). Cells were harvested 48 h after irradiation (6 Gy). (**F**–**H**) ACT-qPCR signals for the histone modifications H3K4me1 (**F**), H3K27ac (**G**), and H3K27me3 (**H**) at the EGR1-binding sites of *DGKA* enhancer (EGR1_BD1 and EGR1_BD2), gene promoter (Promo) and a downstream control region (Ctrl) in HEK293T and cumate-treated dCas9-VPR-TET3-expressing cells. Results from (**C**) and (**E**–**H**) are shown as mean ± SEM from three biological replicates. Statistical significance (* *p* < 0.05 and ** *p* < 0.01) was determined by one-way ANOVA analysis followed by Tukey’s test (**C**) or one-tailed Student’s *t*-test (**E**–**H**).

**Figure 2 cancers-13-02455-f002:**
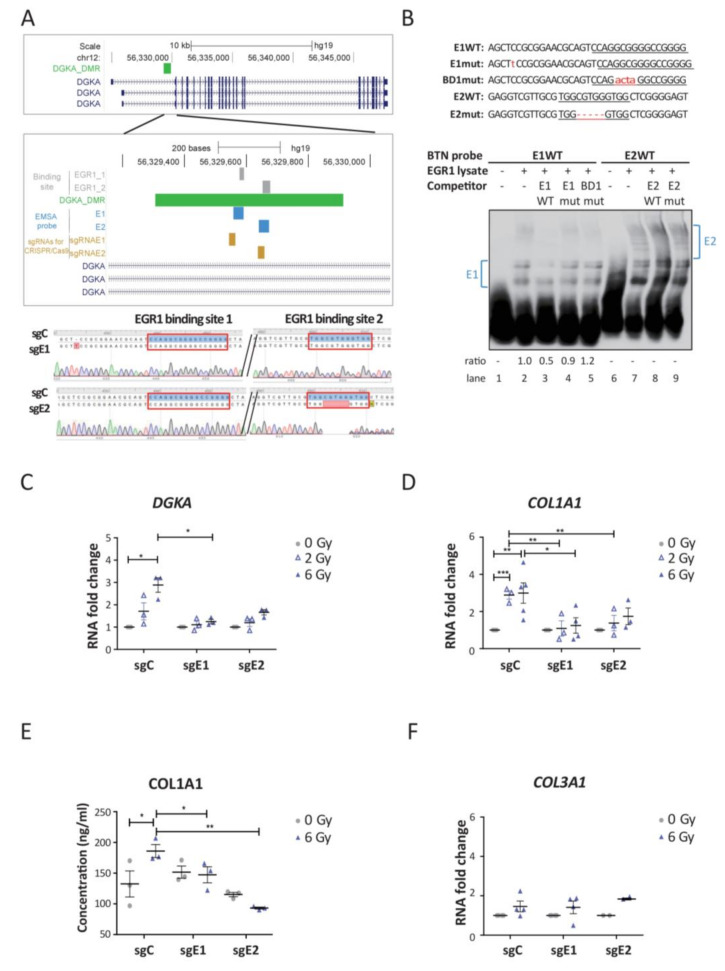
Loss of EGR1-binding at the *DGKA* enhancer suppresses the induction of *DGKA* and COL1A1 after irradiation. (**A**) Upper panel: map of the interrogated *DGKA* enhancer region in relation to DGKA transcripts. Zoom-in: indicates the two EGR1-binding sites (EGR1_1 and EGR1_2, gray), the differentially methylated region (DMR), regions for EMSA probes (E1 and E2, blue), two sgRNA targeting sites (sgRNAE1 and sgRNAE2; khaki), and *DGKA* transcripts (navy). Lower panel: DNA sequence analysis covering the two EGR1-binding sites (red boxes) in the *DGKA* enhancer. sgC indicates the sequence of control (wild type) sgRNA-treated cells. sgE1 and sgE2 indicate cells targeted with sgRNAE1 and sgRNAE2 at the first and the second EGR1-binding site, respectively. Mutations are highlighted in red. (**B**) Upper panel: sequences of probes used in EMSA. E1WT and E2WT probes contain the wild type sequence covering the first and the second EGR1-binding sites. E1mut probe contains a one-base pair insertion before the first EGR1-binding site as observed in sgE1 cells. BD1mut probe contains a GGCG to ACTA mutation directly in the first EGR1-binding site. E2mut probe contains a five-base pair deletion in the second EGR1-binding site as observed in sgE2 cells. Lower panel: EGR1 DNA-binding activity at the TF-binding sites measured by EMSA. Ratios indicated are based on the E1 or E2 band shifts highlighted with blue brackets. (**C**,**D**) Relative mRNA expression of *DGKA* (**C**) and *COL1A1* (**D**) in edited (sgE1, sgE2) and control cells (sgC). Cells were harvested 48 h after irradiation with 2 or 6 Gy. (**E**) COL1A1 secretion was measured by ELISA. Culture medium was harvested 72 h after irradiation with 6 Gy. (**F**) Relative mRNA expression levels of *COL3A1* in the indicated cells. Cells were harvested 48 h after irradiation with 2 or 6 Gy. Data from (**C**–**F**) are presented as mean± SEM from three (**C**,**E**), at least three (**D**) or at least two (**F**) biological replicates. Statistical significance (* *p* < 0.05, ** *p* < 0.01, and *** *p* < 0.001) was determined by one-way ANOVA analysis followed by Bonferroni’s multiple comparison test (**C**,**D**) or one-tailed Student’s *t*-test (**E**,**F**).

**Figure 3 cancers-13-02455-f003:**
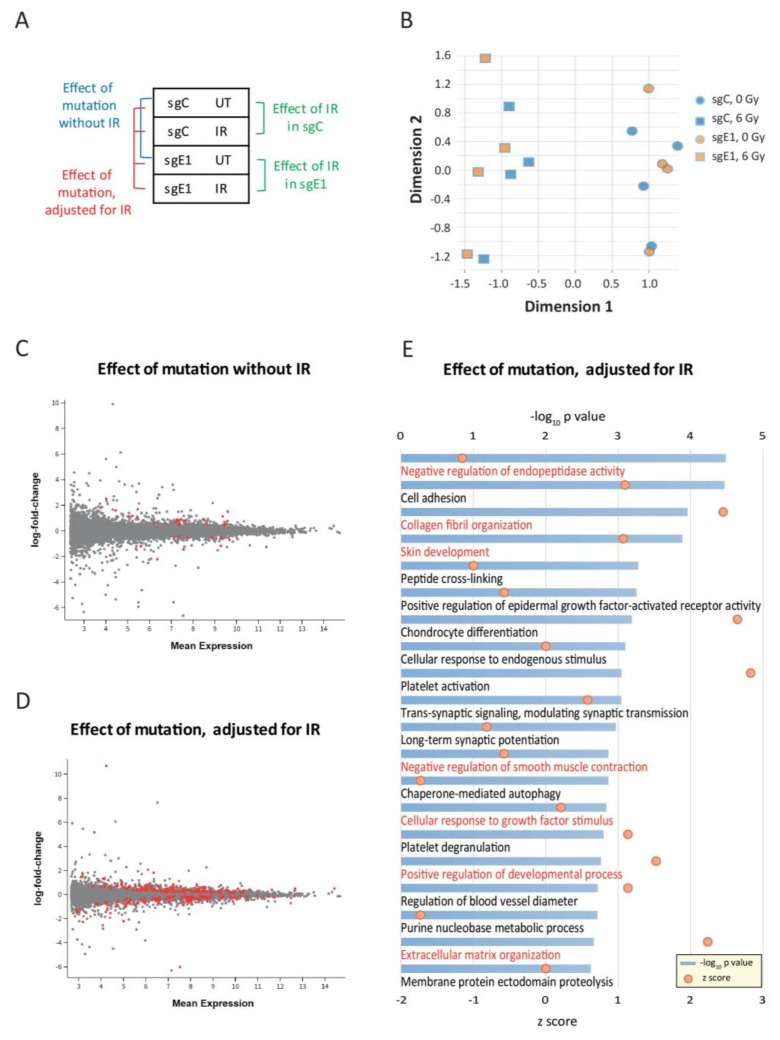
Genome-wide effects induced by loss of EGR1-binding at the *DGKA* enhancer. (**A**) Schematic diagram of the comparisons between control cells (sgC) and sgE1 cells with loss of EGR1-binding at the *DGKA* enhancer. (**B**) Multi-dimensional scaling (MDS) plot comparing all genes expressed in sgC and sgE1 cells with or without irradiation treatment (6 Gy). (**C**) Plot presenting the differentially expressed genes (DEGs) obtained from the comparison of sgC and sgE1 cells without irradiation. (**D**) Plot presenting the DEGs of sgC and sgE1 cells with and without irradiation, adjusted for irradiation. Mean expression values are log-normalized. Genes with adjusted *p*-values (p-adj) < 0.05 are highlighted in red. (**E**) Gene ontology (GO) analysis of DEGs from the comparison presented in Figure 3D. To identify the effect of the binding site mutation, statistical tests were adjusted for irradiation. Significance (-log_10_(*p* value)) of GO terms is presented by blue bars, and z scores are shown with red dots. Biological functions and pathways potentially related to fibroblast activation and fibrosis development are highlighted in red.

**Figure 4 cancers-13-02455-f004:**
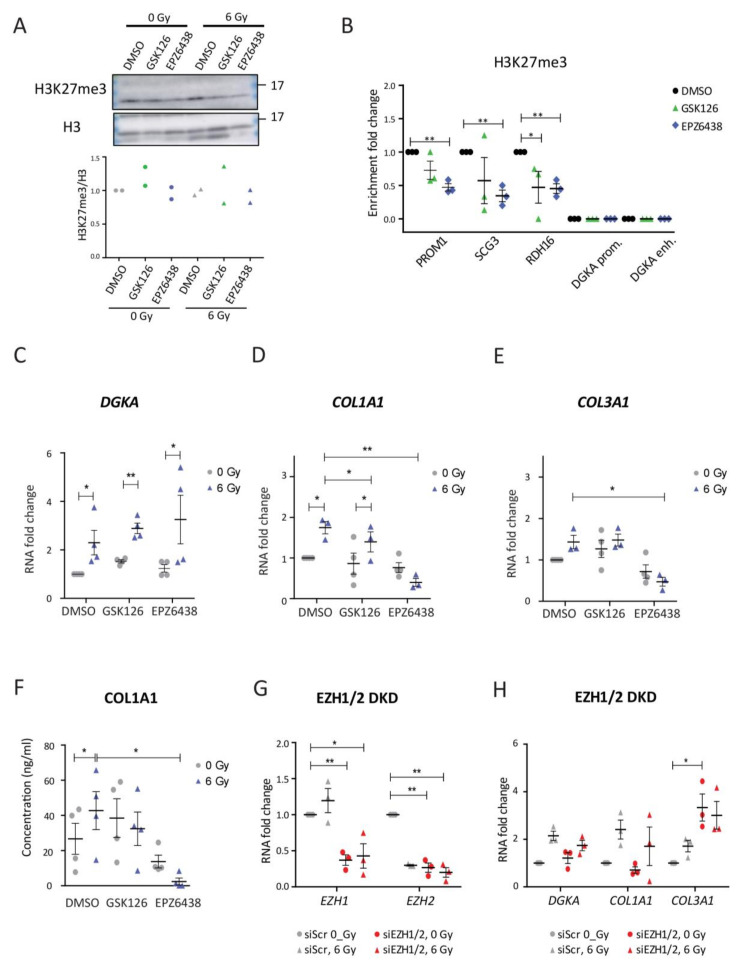
Treatment with EZH1/2 inhibitors did not affect radiation-induced *DGKA* expression and suppressed the induction of a pro-fibrotic response. (**A**) Nuclear H3K27me3 expression in BJ cells treated with EZH2 inhibitors. (**B**) Quantitative ACT-seq signals for H3K27me3 enrichment at the indicated loci in BJ cells. (**C**–**E**) Relative mRNA expression of *DGKA* (**C**), *COL1A1* (**D**), and *COL3A1* (**E**) in BJ cells. For (**A**–**E**), cells were pre-treated with DMSO, GSK126 (5 μM), or EPZ6438 (50 μM) for 48 h, irradiated with 6 Gy, and harvested after an additional 48 h with concurrent drug treatment. (**F**) Secreted COL1A1 was measured by ELISA. Cells were pre-treated with DMSO, GSK126 (5 μM), or EPZ6438 (50 μM) for 48 h, irradiated with 6 Gy, and the conditioned medium was harvested after an additional 72 h with concurrent drug treatment. (**G**,**H**) Relative mRNA expression of *EZH1* and *EZH2* (**G**) as well as *DGKA*, *COL1A1,* and *COL3A1* (**H**) in BJ cells after double-knockdown of (DKD) of EZH1 and EZH2. Cells were pre-treated with siRNAs directed against both enzymes for 48 h, irradiated with 6 Gy, and harvested 48 h later. Statistical data are presented as mean ± SEM from three (**B**–**G**) or four (**F**) biological replicates. Statistical significance (* *p* < 0.05 and ** *p* < 0.01) was determined by one-tailed Student’s *t*-test.

**Figure 5 cancers-13-02455-f005:**
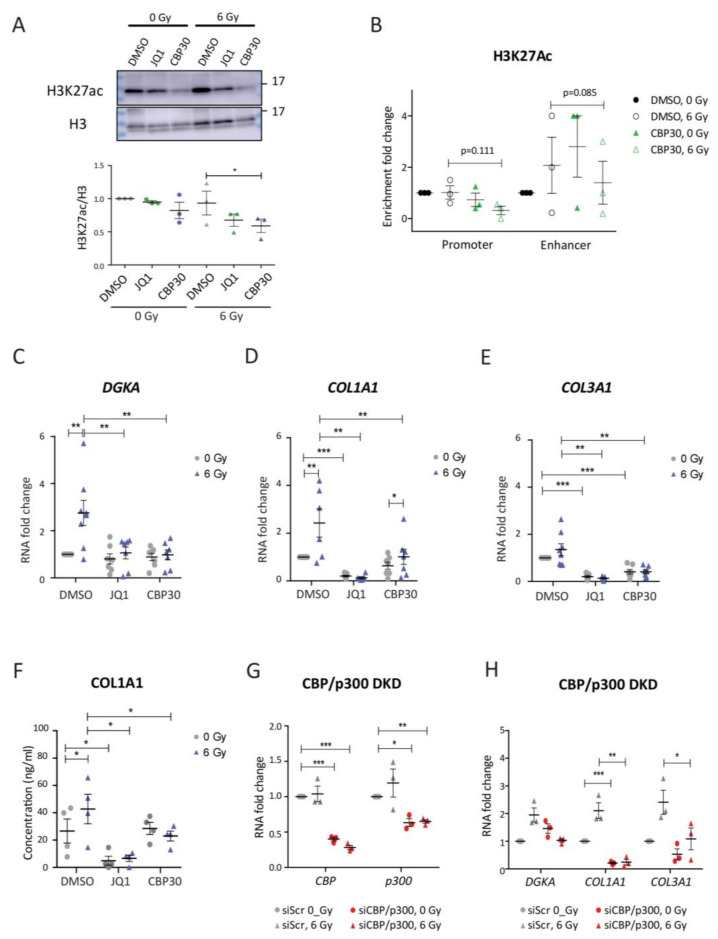
Bromodomain inhibitors reduce radiation-induced *DGKA* and pro-fibrotic marker expression in BJ cells. (**A**) Nuclear H3K27ac expression in BJ cells treated with bromodomain inhibitors. (**B**) Quantitative ACT-seq signals for H3K27ac enrichment at the *DGKA* locus in BJ cells. (**C**–**E**) Relative mRNA expression of *DGKA* (**C**), *COL1A1* (**D**), and *COL3A1* (**E**) in BJ cells. For (**A**–**E**), cells were pre-treated with DMSO, CBP (10 μM), or JQ1 (5 μM) for 48 h, irradiated with 6 Gy, and analyzed after an additional 48 h with concurrent drug treatment. (**F**) Secreted COL1A1 was measured by ELISA. Cells were pre-treated with DMSO, CBP (10 μM), or JQ1 (5 μM) for 48 h, irradiated with 6 Gy, and the conditioned medium was harvested after a further 72 h with concurrent drug treatment. (**G**,**H**) Relative mRNA expression of *CBP* and *p300* (**G**) as well as *DGKA*, *COL1A1,* and *COL3A1* (**H**) in BJ cells after silencing *CBP* and *p300*. Cells were pre-treated with siRNAs directed against both enzymes for 48 h, irradiated with 6 Gy, and harvested after an additional 48 h. Statistical data are presented as mean ± SEM from three (**A**,**B**,**G**,**H**), six (**C**–**E**) or four (**F**) biological replicates. Statistical significance (* *p* < 0.05, ** *p* < 0.01, and *** *p* < 0.001) was determined by one-tailed Student’s *t*-test.

**Figure 6 cancers-13-02455-f006:**
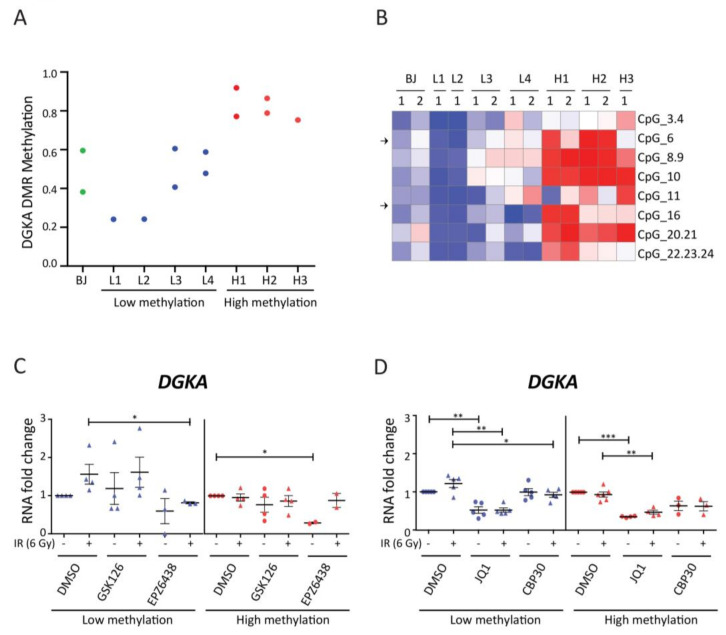
Effects of epigenetic inhibitors in human dermal fibroblasts with low or high DNA methylation at the *DGKA* enhancer: (**A**) Methylation average across informative CpGs located in the *DGKA* enhancer in BJ and normal human dermal fibroblasts (NHDFs, L1-4 and H1-3). Methylation was measured as β-values by EpiTYPER technology. (**B**) Heatmap for DNA methylation of all CpG sites measured in BJ and NHDFs. For each cell type, one or two replicates (1,2) are shown. Arrows indicate the location of the EGR1-binding motifs. The first EGR1-binding site (EGR1_1) covers CpG_6, and the second binding site (EGR1_2) covers CpG_14, which is located between CpG_11 and CpG_16 and not detected by EpiTYPER. For (**A**,**B**), data are depicted from one or two experiment in each single NHDF. (C-D) Cells were pre-treated with EZH2 inhibitors (5 μM GSK126 or 50 μM GSK126) (**C**) or bromodomain inhibitors (10 μM CBP30 or 5 μM JQ1) (**D**), irradiated with 6 Gy, and incubated for another 48 h with concurrent drug treatment before harvest. Data points summarize effects for both fibroblast groups and include at least duplicate experiments. Statistical data are presented as mean ± SEM, and statistical significance (* *p* < 0.05, ** *p* < 0.01, and *** *p* < 0.001) was determined by one-tailed Student’s *t*-test.

## Data Availability

The data presented in this study are available in Appendix A here. The RNAseq and ACTseq data have been deposited to the European Nucleotide Archive and assigned to the identifier PRJEB43383.

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
