# Peer review of "Epigenetic Modulation of Radiation-Induced Diacylglycerol Kinase Alpha Expression Prevents Pro-Fibrotic Fibroblast Response"

_cancers, 2021, doi:10.3390/cancers13102455_

Round 1

Reviewer 1 Report

This is an extremely lengthy paper with a huge amount of data. The experimental methods are very good.  There are several comments, which may assist the authors in minor revisions if they choose.

Central focus to this manuscript is that differential DNA methylation enhancer of diacylglycerol kinase alpha (DGKA) dermal fibroblasts is associated with irradiation-induced fibrosis. There are many scientific approaches to understanding fibrosis, and this is a novel approach. Since the field is essentially wide open for novel and interesting approaches to understanding radiation fibrosis, this manuscript represents a useful and positive addition to the literature.

Specific Comments:

Molecular biology and the data in Figure 1 is outstanding.

The method section describes work with cell line and the molecular biology to follow. The authors may wish to reference their work or that of others on animal model systems in which these approaches should at some point be confirmed.

The molecular biology data in Figures 2 and 3 is excellent. The data in Figure 4 with respect to EZH2 inhibitors is very important and quite well presented.

Figure 5 shows data on Bromodomain inhibitors that effectively reduce irradiation induced DGKA and is extended in Figure 6.

The discussion section may be modified after inserting some statements about in vivo correlations of their data with cell lines.

The supplemental figures are extremely helpful, as are the data sets in the raw data presentation and the reviewers, who are interested in the background information may find this useful.

Reviewer 2 Report

Thank you for the opportunity to review “Epigenetic modulation of radiation-induced DGKA expression prevents pro-fibrotic fibroblast response” by Liu et al. Authors build on their previous work on the role of diacylglycerol kinase alpha (DGKA) in radiation-induced fibrosis. The current manuscript uses multiple cell line models such as HEK293T, BJ fibroblasts and patient-derived fibroblasts and shows that the editing of the DGKA enhancer using genomic and epigenomic tools results in induction of the DGKA expression. Demethylation of the DGKA enhancer increases the DGKA expression. Mutating EGR1-binding sites using CRISPR-Cas9 resulted in a reduced radiation-induced DGKA expression. Pharmacological bromodomain inhibition also reduced radiation-induced DGKA expression and expression of gene associated with fibrosis. Authors suggested these bromodomain inhibitors as a novel treatment for alleviating fibrosis brought on by radiation.

The idea to characterize the regulation of DGKA expression in radiation-induced fibrosis is important and fills a gap in knowledge in the field. The manuscript is generally well written although some parts of it are presented in a confusing manner regarding which I have some comments.

Main points

In Fig. 1B – the binding of EGR1 also gets reduced when bound to unmethylated E2 probe compared to methylated E2 probe because authors say: not altered when incubated with the methylated E2 (E2me) probe”. Is this effect not reproducible? Authors should explain this.

Authors should explain why H3K27me3 enrichment at BD1 and BD2 sites did not result in repression of DGKA expression. Was it overcompensated by an enrichment in H3K4me1 and H3K27ac?

Authors state that COL3A1 is weakly induced upon irradiation. Is this a significant increase? If so then statistics should be included in the figure 2F.

In Fig 4A SEM should not be used for 2 replicates. Authors should add a third replicate or remove SEM error bars from their current figure.

Authors should discuss the potential implications of the increased variability H3K27Ac after JQ1 and CBP30 treatments following irradiation as a response mechanism by the cell to irradiation after blocking histone acetyltransferases. It may indicate cell plasticity and a range of transition states.

Page 20 says that “CBP30 significantly suppressed the radiation-induced COL1A1 and COL3A1 expression, especially in low methylated NHDFs” but this trend is shown for the high methylation group so the authors should correct this error and comment in their discussion why JQ1 and CBP30 show a difference in attenuation of DGKA expression in low vs high methylation primary fibroblasts.

Minor points

Cartoon illustrations in panel A and B are redundant and should be combined into one panel that should clearly mark E1 and E2 sites in relation to the gene and enhancer sequence and DMR. The DGKA gene in panel A and DGKA DMR in the enhancer in panels A&B are both green but DGKA enhancer itself in panel A is actually yellow. This is confusing because it looks like E1 and E2 sites are within the DGKA gene rather than the enhancer. This should be clarified. Does the entire yellow box in the lower part of panel A depict the extent of the entire DGKA enhancer? Perhaps the upper and lower parts should be connected but lines that fan out from the enhancer otherwise this is confusing.

An illustration of the Cas9 system should be provided in panel C or as a separate panel in Fig. 1 because this information is confusing when added to the legend of Fig. 1A.

Fig. 1D legend should explain what do numbers 1, 2, 3 mean above the heatmap. Are these experimental replicates?

Reviewer 3 Report

The manuscript of Chun-Shan Liu and colleagues entitled "Epigenetic modulation of radiation-induced DGKA expression prevents pro-fibrotic fibroblast response" is very well designed and clear. The authors demonstrated that epigenomic changes of DGKA expression may be a potential target to treat or prevent fibrosis induced by radiotherapy. The manuscript is well written and may be accepted in this form. However, I suggest to the authors to do a figure/ scheme to summarize the data and facilitate the understanding of the results.

Author Response

We thank the reviewer for acknowledging our approach. To make our procedure even clearer, we created a graphical scheme to explain our concept, and added it to the manuscript as Figure S7.

Reviewer 4 Report

This is a well-written and interesting manuscript in which Liu and colleagues described the effects of genetic and epigenetic perturbations of the DGKA enhancer in the induction of a pro-fibrotic pathway of gene expression after irradiation of cells, resembling what happens during radiotherapy treatments. Moreover, the authors identified bromodomain inhibitors as major modulators of DGKA expression when its enhancer is in a pro-fibrotic, low methylated state.

Few comments:

- Regarding the induction of demethylation of the DGKA enhancer in HEK293 cells by dCas9-TET3, loss of methylation is not complete, neither after cumate treatment (Fig. 1C,D). Is this reduced methylation sufficient to allow EGR1 binding to the region? Is it possible that cell irradiation, a part to increase histone modifications, also increase DNA demethylation of the enhancer region? In other words, how is the DNA methylation of EGR1 binding sites in dCas9-TET3+cumate cells after irradiation? Moreover, are the expression levels of EGR1 comparable between wild-type and dCas9-TET3+cumate mutant cells, both before and after irradiation?

- The E1mut probe contains one T insertion only compared to the wild-type enhancer sequence and it does not fall into the first EGR1 binding site (Fig. 2A). However, this mutant fibroblast cell line showed the strongest effect on DGKA andCOL1A1 expression levels after irradiation (Fig. 2C,D) and indeed it has been also chosen for RNA sequencing analysis. Do the authors have an hypothesis on the mechanism by which E1mut can affect EGR1 binding at the DGKA enhancer?

- Figure 1D: Please, illustrate the position of the 8 analysed  CpGs compared to the EGR1 binding sites.

- Figure 1A: It does not seem that BD1 covers EGR1 and that BD2 covers EGR2.

- Results 3.4: “As already observed (Figure 1C-E)…” may be (Figure “C-E) ?

Round 2

Reviewer 4 Report

I was pleased to see that the authors have addressed my comments and major criticisms. The revised version of the manuscript can now be accepted for publication in Cancers.